# Dmg2Former-AR: Vision Transformers with Adaptive Rescaling for High-Resolution Structural Visual Inspection

**DOI:** 10.3390/s24186007

**Published:** 2024-09-17

**Authors:** Kareem Eltouny, Seyedomid Sajedi, Xiao Liang

**Affiliations:** 1Department of Civil, Structural and Environmental Engineering, University at Buffalo, The State University of New York, Buffalo, NY 14260, USA; kaeltouny@sgh.com (K.E.); ssajedi@thorntontomasetti.com (S.S.); 2Structural Mechanics & Materials Division, Simpson Gumpertz & Heger, Waltham, MA 02451, USA; 3CORE Studio, Thornton Tomasetti, New York, NY 10271, USA; 4Zachry Department of Civil & Environmental Engineering, Texas A&M University, College Station, TX 77840, USA

**Keywords:** structural health monitoring, structural condition assessment, visual inspection, semantic segmentation, vision transformers, deep learning, computer vision, super resolution, downsampling

## Abstract

Developments in drones and imaging hardware technology have opened up countless possibilities for enhancing structural condition assessments and visual inspections. However, processing the inspection images requires considerable work hours, leading to delays in the assessment process. This study presents a semantic segmentation architecture that integrates vision transformers with Laplacian pyramid scaling networks, enabling rapid and accurate pixel-level damage detection. Unlike conventional methods that often lose critical details through resampling or cropping high-resolution images, our approach preserves essential inspection-related information such as microcracks and edges using non-uniform image rescaling networks. This innovation allows for detailed damage identification of high-resolution images while significantly reducing the computational demands. Our main contributions in this study are: (1) proposing two rescaling networks that together allow for processing high-resolution images while significantly reducing the computational demands; and (2) proposing Dmg2Former, a low-resolution segmentation network with a Swin Transformer backbone that leverages the saved computational resources to produce detailed visual inspection masks. We validate our method through a series of experiments on publicly available visual inspection datasets, addressing various tasks such as crack detection and material identification. Finally, we examine the computational efficiency of the adaptive rescalers in terms of multiply–accumulate operations and GPU-memory requirements.

## 1. Introduction

The aging United States infrastructure requires substantial financial investments for repairs or replacement. With 42% of all bridges being at least 50 years old [1], an investment backlog of $132 billion is estimated for the rehabilitation of existing bridges [2]. A similar trend is observed for U.S. dams and waterway structures, with the cost of rehabilitation of the non-federally owned dams rising to $157.5 billion in 2023 [3]. It is evident that the choice of structural condition assessment strategies can influence the safety and economic growth of communities, as they directly impact the functionality of the constantly deteriorating infrastructure as well as the recovery process of urban areas following a natural hazard.

Traditionally, condition assessments are performed through field inspections with a preliminary visual evaluation and further non-destructive evaluation tests. Significant work hours and logistics are often utilized during this process, making it expensive, time-consuming, and unsafe for inspectors in many situations. In an effort to improve the efficiency and reliability of structural inspections, a noteworthy amount of research has been conducted to streamline the visual inspection process.

Vision-based structural health monitoring (SHM) and automated visual inspections rely on visual footage of the structure to identify structural materials and components or detect surface damage (e.g., cracks, spalling, and rust). By benefiting from extensive research in computer vision technology, vision-based systems can process massive amounts of inspection images, such as those collected from a large urban area hit by a catastrophic event. Therefore, vision-based SHM can significantly cut down the time needed for condition assessments. Traditional crack and spalling detection methods employ morphological operations, edge detection algorithms, binarization, and other image processing techniques [4,5]. More recently, machine-learning algorithms have been the preferred option for analyzing inspection images [6].

Recent advancements in deep learning, coupled with improvements in computing and sensing technologies, have noticeably accelerated SHM research [7]. Wireless accelerometers have seen improvements in accuracy, efficiency, and cost, allowing the deployment of advanced SHM systems leveraging large sensor arrays [8,9]. In turn, numerous methodologies were proposed in recent years that leverage machine-learning methods to identify structural damage based on vibration measurements [10,11,12]. Advances in strain sensing, especially radio frequency identification (RFID)-based wireless sensors, also allow for remote monitoring of structural components’ deformation as part of condition assessment frameworks [13]. RFID sensors have been used for crack width measurements [14], concrete moisture monitoring [15], and long-distance structural health monitoring [16].

Sensing technology breakthroughs are also notable in vision-based SHM. Pipeline closed circuit television video (CCTV) inspections are increasingly being used to remotely identify pipe degradation and distress [17,18,19]. The cost and efficiency of camera-equipped unmanned aerial vehicles (UAVs) have led to the wide adoption of their use in structural visual inspections [20,21]. Underwater robots [22], infrared cameras [23], LiDAR [24], and depth sensors [25] have also been adopted for performing visual inspections. The variety of visual-sensing technologies allows the fast collection of vast amounts of high-fidelity visual data with minimal risk to the safety of inspection personnel. However, manually inspecting thousands of collected images for structural defects is a time-consuming process. Therefore, developing machine-learning frameworks to precisely and efficiently parse inspection images is of paramount importance.

Multiple studies have focused on detecting building components and materials that can be used to facilitate the inspection process or for UAV motion planning. Examples of methods proposed include object detection [26,27] and semantic segmentation [28,29]. The bulk of automated visual inspection research focused on detecting various types of structural damage. For concrete cracks, some methods predicted crack regions using multi-stage or single-stage object detection networks, such as YOLOv2 [30] or YOLOv8 [31]. Other methods relied on deep-learning semantic segmentation models to create crack maps including U-Net [32], DenseNet [29], stacked convolutional autoencoders [33], and deep CNNs [34,35,36]. Xu et al. [37] proposed a lightweight crack segmentation model using DeepLabV3+ with a MobileNetV2 backbone. Choi et al. [38] combined a CNN-based image classifier and clustering to identify the percentage of crack areas in thermal images. Sohaib et al. [39] used an ensemble of YOLOv8 networks with various sizes to detect and segment concrete cracks. Yang et al. [40] used K-Net to detect structural surface defects from UAV-acquired images and a texture-mapping method to map the defects to a BIM model. The attention mechanism has also been widely used in recent years in crack segmentation models. Hang et al. [41] relied on feature compression and channel attention modules to improve crack segmentation performance. Yu et al. [42] proposed a U-Net-like architecture with attention-based skip connections for the crack segmentation of a nuclear containment structure. Other examples of damage identification include rail surface defects [43], building façades [44], fatigue cracks, spalling, and corrosion [45,46,47].

Super-resolution architectures have also been used in crack detection. Bae et al. proposed SrcNet for crack detection based on UAV images, which initially generates a super-resolution version of the image before processing it through a segmentation model [48]. Similarly, Xiang et al. used a super-resolution reconstruction network to refine fuzzy UAV images before performing crack segmentation [49]. Kim et al. proposed using SRGAN for enhancing input images to predict refined concrete crack masks [50].

As is evident from the recent literature, using off-the-shelf deep-learning algorithms for developing vision-based SHM is a common approach. However, given their critical nature, structural inspections demand high precision and accuracy. Even though many visual data acquisition devices can capture high-resolution images, limited resources often dictate significant downsizing of images to allow the use of state-of-the-art deep-learning models. This is especially necessary for vision transformers (ViT), which are becoming increasingly expensive in terms of computational demands. Additionally, many visual inspection tasks, such as UAV motion planning, require efficient models capable of real-time inference. Recently, two models were proposed to analyze high-resolution inspection images based on two strategies, each suited to different visual inspection tasks [51]. Despite these efforts, no single model fully met the diverse requirements of various visual inspection tasks, and there were limited attempts to address these challenges.

In this study, we develop a unified framework for high-resolution visual inspection that can strike a balance between prediction quality and computational efficiency. Uniform downsizing of images, commonly performed when resources are limited, can distort the original image and cause a loss of fine details. We propose Dmg2Former with adaptive rescaling (Dmg2Former-AR), a transformer-based segmentation model paired with cascaded sub-pixel convolution rescaling networks for analyzing visual inspection images. Dmg2Former-AR brings the ViT technology into the field of vision-based SHM with minimal increase in computational costs. Our main contributions in this study are (1) proposing two rescaling networks that together allow for processing high-resolution images while significantly reducing the computational demands; and (2) proposing Dmg2Former, a low-resolution segmentation network with a Swin Transformer backbone that leverages saved computational resources to produce detailed visual inspection masks. We evaluate our proposed model for two tasks: materials segmentation and crack damage segmentation.

The remainder of this paper is organized as follows. The following section describes the Dmg2Former and the adaptive downscaler and upscaler architectures. Following this discussion, we dedicate a section to describing the two case studies and detailing the implementation of our models. Then, we present the results of the evaluation metrics used in testing. Finally, we provide a summary, conclusions, and future research directions for this study.

## 2. Architecture Design

Using off-the-shelf deep-learning models for the segmentation of high-resolution images can be significantly inefficient and is often constrained by GPU-memory limitations. There are two common strategies to deal with high-resolution images for semantic segmentation. The first is downscaling the input image using interpolation-based resampling techniques to reduce the computational overhead during inference and training (Figure 1a). If needed, a high-resolution mask can also be obtained via nearest-neighbor or interpolation-based resampling. This method, however, suffers from loss or distortion in fine details, which may be acceptable in some computer vision applications, but it can drastically jeopardize the visual inspection process as details such as thin cracks and edges can hold valuable information.

The second method, which is more common for vision-based civil engineering applications, is cropping the image into smaller-sized patches to be fed to the deep-learning model (Figure 1b). The predicted mask patches are used to stitch the full-resolution image back together. While the local details remain primarily intact during cropping, the global contextual information of the image can be lost in the process. It is also worth mentioning that while this procedure makes the computational burden significantly smaller than if we used the full-resolution image as-is, it remains more computationally expensive than resampling the image because the model must make multiple forward passes to fully construct the mask.

We strive to deliver high-resolution segmentation of visual inspection images while upholding the efficiency of low-resolution segmentation models and adhering to the GPU memory constraints. To this end, we propose two neural networks that can adaptively and efficiently downscale the images and upscale the masks while preserving the fine details necessary for the visual inspection procedure (Figure 1c).

The overall framework of Dmg2Former-AR is shown in Figure 2. The architecture consists of three main modules: a downsampling encoder, an inner segmentation model, and an upsampling decoder. When fed with an HR image, the downsampling decoder adaptively produces an LR version of the image, which preserves the essential information in the limited number of pixels of the LR image at the expense of less-valued details. Then, the transformer-based segmentation model efficiently predicts an LR mask given the LR image input at significantly lower computational costs. Finally, the upsampling decoder upscales the LR mask and brings it as close as possible to the true HR mask.

### 2.1. Adaptive Rescaling

When compressing visual inspection images, it is understood that certain pixels hold more significance than others. Consequently, adaptive resamplers are designed to learn how to sample valuable pixels more frequently during the sampling process, addressing the limitations of uniform sampling. They are adaptive in that, unlike uniform resamplers, they adjust the downsampling or upsampling grids according to the contents of the input image or mask. With this in mind, the downsampling encoder and upsampling decoder take inspiration from advances made in real-time image super-resolution models, including the efficient sub-pixel convolution network [53], residual super-resolution networks [54,55,56], and deep Laplacian pyramid networks [57,58].

The downsampling encoder and upsampling decoder are called Laplacian Subpixel Convolutional Network (LapSCN) and Laplacian Desubpixel Convolutional Network (LapDCN), respectively (Figure 3). These networks can progressively rescale images or masks based on a cascade of subpixel convolutional modules. LapSCN has a mask reconstruction stream comprising a set of feature extraction modules with residual learning, while LapDCN has an image deconstruction stream accompanied by its own feature extraction modules. At each stage, the feature extraction modules process the input data to predict lower- or upper-scale residuals. These residuals are then added to the resized data in the deconstruction or reconstruction stream, refining the output. The output of these streams can be used as the final resized output or can undergo further scaling by re-entering the feature extraction modules.

Each feature extraction module in LapSCN is a subpixel convolutional module that predicts residuals for masks upscaled by a factor of two (right side of Figure 3). The module consists of three convolutional blocks, which include a convolutional layer (CN), batch normalization (BN) [59], and rectified linear unit activation (ReLU), followed by a pixel shuffler operation. The pixel shuffle layer [53] aggregates the low-resolution (*w*/2, *h*/2, *n* × 4) feature maps to form twice-upscaled masks (*w*, *h*, *n*), where *n* represents the number of classes (Figure 4). Pixel shuffle was shown to improve the performance of super-resolution networks compared to other upsampling convolution layers. It is highly efficient, reduces checkerboard artifacts, and allows the network to directly predict the high-resolution pixels without intermediate upsampling steps [53].

The reconstruction stream is a global skip connection that helps stabilize the network training and facilitates convergence. In the deep Laplacian pyramid super-resolution network [57], the bicubic interpolation was used to upscale the images to the target resolution in the reconstruction branch before adding the residuals. Instead, we use repeat upscaling, which is found to be significantly faster than interpolation upsampling methods [56,60]. Repeat upscaling is a special case of nearest neighbor interpolation. It repeats the low-resolution feature map along the channel dimension and is added to the residuals before the pixel shuffle operation. While conceptually, pixel shuffle of mask repeats should produce the same result as a nearest neighbor upscaling operation, experiments have shown that it is computationally cheaper to perform [56,60].

On the other side, LapDCN has desubpixel convolutional modules that start with downscaling the input image using a pixel unshuffle operation, a reverse of the pixel shuffle operation, followed by three convolutional blocks, that ends with a sigmoid activation (left side of Figure 3). By rearranging the pixels along the channels’ axis, the pixel unshuffle layer significantly reduces the computational cost without loss of information (Figure 4). Pixel unshuffle transforms a high-resolution image (*w*, *h*, *c*) to a twice-downscaled version (*w*/2, *h*/2, *c* × 4) with four times more channels. At the first convolutional layer, the computational savings obtained from downscaling the image are offset by the expenses due to the expansion of channels. However, substantial net computational savings are achieved in the subsequent convolutional operations as a result of the downscaling operation. Finally, the input image is downsized in the image deconstruction stream using adaptive average pooling before adding the residuals from the feature extraction module. Details regarding the filters of the convolutional layers of the feature extraction modules of both LapDCN and LapSCN are presented in Table 1.

### 2.2. Dmg2Former

The internal segmentation model, Dmg2Former, uses a Swin Transformer backbone [52]. Originally used in natural language processing [61], transformers have seen success in several computer vision tasks and have since achieved state-of-the-art results on well-known benchmarks, including ImageNet and ADE20k [62,63,64]. The Swin Transformer introduced a multi-stage approach for vision transformers. It relies on patch merging operations to provide hierarchical feature maps, offering the possibility of using it as a backbone for well-established segmentation decoders. With a few adjustments, we used a Swin Transformer Base (Swin-B) model. Our decoder is inspired by U-Net++ [65], a U-Net variant explicitly developed for medical imaging segmentation. Compared to U-Net, U-Net++ adds complex, dense convolutional blocks to the skip connections to bridge the semantic gap between the encoder and decoder. We modified the U-Net++ architecture implemented by Iakubovskii, P. [66] to be compatible with the Swin Transformer encoder, making it four stages in depth instead of the original five. In summary, Dmg2Former is an encoder–decoder segmentation model with a base Swin Transformer backbone adapted with a custom-built decoder with inspirations from U-Net++. The architecture of Dmg2Former is shown in Figure 2.

## 3. Case Studies

### 3.1. Datasets Description

We used two visual inspection datasets for validation purposes. The first is the material segmentation dataset, a publicly available dataset containing 3817 images created from bridge inspection reports of the Virginia Department of Transportation [67] with pixel-level annotation of three structural inspection materials: concrete, steel, and metal decking. The image resolution varies significantly with resolutions ranging from 258 × 37 to 5184 × 3456. With a sizable pool of high-resolution images, the dataset is a good candidate for testing our framework. The dataset is split by the original authors into 3436 samples for training and 381 samples for testing. We further set aside 352 samples from the training dataset for validation during network training (approximately 10% of the training set).

The other dataset is the Concrete Crack Conglomerate dataset, a publicly available dataset containing 10,995 images of cracks with pixel-level annotation obtained from multiple other datasets, shown in Table 2 [67,68]. As with the majority of the crack segmentation datasets, this dataset suffers from high class imbalance due to crack pixels occupying 2.8% of all pixels. All images in the dataset are 448 × 448 in size. Therefore, we train the resizers to upsize the segmentation model input and output resolution up to that resolution (factors of either two or four). Nevertheless, the model can be used to infer images that are larger in size due to the scaling capabilities of its Laplacian pyramids. Regarding the training/testing data split, the dataset was already split into a 90/10 ratio for training and testing by the original authors. We also reserve 10% from the training set for validation purposes.

### 3.2. Implementation

One of the main challenges in training the model to provide pixel-wise labeling of cracks is that the size of the cracks is insignificant compared to the background. Without taking measures, the model may classify all pixels as background and would still achieve sufficiently high accuracy. We use various training techniques to handle this class imbalance, including focal loss, regularization, and data augmentation.

The models were built and trained using the PyTorch library (v2.0.1) [73] and a workstation equipped with Intel^®^ Core i9-13900k CPU (Intel, Santa Clara, CA, USA) and an Nvidia RTX 4090 GPU (Nvidia, Santa Clara, CA, USA). We used the focal loss [74] as our loss function. Focal loss can help reduce the effects of class imbalance by applying a modulating term to the cross-entropy loss to focus learning on hard, misclassified examples. The focal loss is defined as:(1)$$FL\left(p\right)=\left\{\begin{array}{c} -\alpha {\left(1-p\right)}^{\gamma }\log\left(p\right),\\ -\left(1-\alpha \right)p^{\gamma }\log\left(1-p\right), \end{array}\begin{array}{c} y=1\\ otherwise \end{array}\right.$$
where α is the balancing parameter and γ is the modulating parameter. Based on our experiments, the focal loss seems to improve the prediction accuracy compared to the cross-entropy loss. In this study, we use α = 0.6 for the crack segmentation tasks, α = 0.25 for other tasks, and γ = 2.0 for all visual inspection tasks.

For the optimizer, we use Adam with decoupled weight decay (AdamW) [75] with an initial learning rate of 1 × 10^−4^ and a weight decay of 1 × 10^−6^. We also use cosine annealing with a warm restarts learning rate scheduler and a specified minimum learning rate of 1 × 10^−7^. All models were trained for 255 epochs, and the selected checkpoint corresponds to weights producing the maximum intersection-over-union (IoU) of the validation set. Furthermore, we initialized the parameters of the Swin Transformer backbone using the ImageNet [76] training parameters made available by the original authors whenever possible, mainly when we use an input/output resolution of 224 × 224.

In addition, we used data augmentation techniques during training, adopted from the Albumentations library [77], comprising multiple image transforms such as random horizontal flip, zoom, and perspective. It also included random color manipulations such as brightness, gamma, and saturation. Data augmentation proved to boost the model’s robustness to unseen data despite slowing the training process. Additionally, and to facilitate training, each channel of the input images was standardized based on the channel mean and standard deviation of the training images set. This means that the training images dataset would have a zero mean and a unit variance.

## 4. Results

### 4.1. Material Segmentation

We investigated three Dmg2Former architectures built for three input sizes: 112 × 112, 224 × 224, and 448 × 448, having all parameters randomly initialized. Additionally, we trained another set of Dmg2Former models with an input size of 224 × 224 and had the Swin encoder pre-trained with the ImageNet dataset. We paired the Dmg2Former versions with 112 and 224 input sizes with interpolation-type resizers (nearest-neighbor; Dmg2Former-NN) and our proposed adaptive resizers (Dmg2Former-AR), with factors of upscaling starting from a factor of two (2×) and up to a factor of eight (8×). Dmg2Former, Dmg2Former-NN, and Dmg2Former-AR, built using the exact internal resolution, share the same internal segmentation model. However, Dmg2Former-AR models get further fine-tuned to optimize the adaptive resizers.

It is worth noting that to implement Dmg2Former for an input size of 112, the Swin encoder had to be slightly adjusted. As 112 is not divisible by 32, we patch-partitioned using a patch size of 2 × 2, making the number of tokens at the final Swin encoder stage the multiplication of the input width and depth, each divided by 16.

Table 3 shows the average F1-score, IoU, recall, and precision of the examined models on the material segmentation dataset with randomly initialized parameters. The models are sorted in an ascending order based on the image/mask size they can process. The first observation when comparing Dmg2Former at two different input sizes (112 and 224) without any form of resizing is that the model trained on higher resolution images can better predict the segmentation masks, as seen by the improved performance across all metrics. This implies that the higher-resolution images contain additional non-redundant information that is valuable for parsing the visual inspection data.

Another finding is that when using an interpolation-based resampler, such as nearest neighbor, the model performance deteriorated compared to its low-resolution counterpart. For example, at size 112, Dmg2Former produced higher IoU (0.768) compared to Dmg2Former-NN 2× (0.739) and Dmg2Former-NN 4× (0.737). We also observed that the evaluation metrics stagnated or slightly deteriorated as we increased the upscale factor of the model using the nearest-neighbor algorithm. On the other hand, pairing Dmg2Former with adaptive rescalers (Dmg2Former-AR) allowed the model to improve its prediction capabilities and provide a higher-resolution mask. Moreover, as more adaptive down- and up-scalers were added to the same Dmg2Former model, we noticed improvements, though subtle, in the evaluation metrics results (e.g., based on 112-sized Dmg2Former: IoU of Dmg2Former-AR 2× (0.79) vs. Dmg2Former-AR 8× (0.802)). This resulted in the Dmg2Former-AR 4× relying on a 112-sized Dmg2Former outperforming a vanilla 224-sized Dmg2Former model. Additionally, Dmg2Former-AR 8× evaluation metrics showed the capabilities of the adaptive rescales in enabling a model to predict masks that have 64 times the pixels of the original segmentation model’s predictions.

Table 4 shows the class-average evaluation metrics of the 224-sized Dmg2Former with the Swin encoder initialized using ImageNet weights provided by the original authors of Swin Transformers. The ImageNet initialization of the Swin encoder resulted in noticeable improvements across all metrics for all 224-sized models. Overall, using adaptive rescalers has allowed for a slight increase in the evaluation metrics, with Dmg2Former-AR 2× achieving the best results. However, non-trainable rescalers (Dmg2Former-NN 2× and Dmg2Former-NN 4×) slightly deteriorated the model performance compared to the low-resolution segmentation model (Dmg2Former). It is worth noting that the differences in performance are subtle, and for this dataset, the use of interpolation-based resizers should not drastically impact the prediction results. Additionally, class-wise IoU results presented in Table 5 show a balanced class-wise prediction. Figure 5 visualizes example predictions of Dmg2Former-AR 4× and Dmg2Former-NN 4× side-by-side with the ground truth masks.

### 4.2. Crack Segmentation

For crack segmentation, three different sizes of Dmg2Former were trained, two of which were paired with both adaptive and uniform resizers. As seen in Table 6, the improvements of the adaptive rescaling variants compared to the uniform rescaling ones are more pronounced for the crack segmentation task. This is mainly due to the highly unbalanced crack masks, as fine cracks can be significantly distorted if the image is uniformly resized, making it challenging to predict the masks. Figure 6 shows the output of the LapDCN and segmentation module for Dmg2Former-AR 4× at a 112 × 112 resolution along with their Dmg2Former-NN counterparts. As seen in the figure, when applying a uniform resampling algorithm, fine cracks are heavily distorted and may even vanish, especially when the image is radically downscaled. On the other hand, our adaptive downsampler, LapDCN, preserves the information necessary to predict the mask while discarding unnecessary pixels. In order to achieve this, the downsampler oversamples the cracks and other valuable pixels, as observed in Figure 6. In addition to that, our adaptive upsampler refines the predicted masks, such as resolving crack discontinuities, and upscales them to the desired higher resolution.

We also used the ImageNet pre-trained weights for the Swin encoder to improve the evaluation metrics. Three models sharing the same architecture of the internal segmentation model were trained and compared based on the evaluation metrics (Table 7). Using the same notation, the Dmg2Former model is the low-resolution segmentation model (224 × 224), while Dmg2Former-NN 2× and Dmg2Former-AR 2× are the upscaled variants when using interpolation-based resizers and adaptive resizers, respectively (448 × 448). Similar observations to the results of the materials segmentation can be made in Table 7.

Looking at the visualization of the models’ predictions for sample test images in Figure 7, we observe the following. First, Dmg2Former-AR can provide pixel-level crack predictions with a reasonable degree of accuracy for different conditions and crack characteristics. Second, the models provide “blank” predictions for non-crack images, indicating that the model does not operate as an edge detector. Finally, and unlike Dmg2Former-AR, the models equipped with interpolation-based resizers (Dmg2Former-NN) suffer significantly for images with fine cracks due to the distortion problem in the resizing operations.

### 4.3. Computational Costs

Table 8 presents the number of parameters, multiple-accumulate (MAC) operations, the size of forward and backward passes, and the inference speed of a single image/mask of various Dmg2Former variants, offering insights into the computational demands and efficiency of each model. These statistics are obtained through the Python package Torchinfo (v1.8.0) [78] for four-class output masks. A MAC operation consists of a multiplication followed by an addition (e.g., y=w·x+b) and therefore correlates with the model’s complexity and inference time. The forward pass size indicates the amount of memory (GPU memory if GPUs are utilized) needed to store a single image in a forward pass. When training, a similar amount of memory is needed to make a backward pass in addition to the amount needed for a forward pass. Training cannot be performed if there is simply not enough memory for a single image to make forward and backward passes, in addition to the size of the parameters themselves. Because batch-training is often used, both with MACs and the memory requirement scale with the batch size, it makes it extremely difficult to train models at higher resolutions.

As seen in Table 8, Dmg2Former-AR offers higher resolution segmentation at 448 and 896 sizes with a slight increase in MACs and memory requirements. The requirements for running Dmg2Former natively at 896 can be prohibitive, as it needs approximately 5 GB in GPU memory for passing single images. Alternatively, using a lower resolution Dmg2Former with adaptive rescalers only needs more than four times less computational demands. The inference speed, presented in frames-per-second (FPS), is estimated on a single machine with an Nvidia 3060 Ti GPU with 8 GB of dedicated memory. By looking at the inference speed, we observed improvements in efficiency and real-time performance using adaptive rescaling as the resolution requirement increases, especially for models with an input size of 896 (10 FPS using Dmg2Former compared to 46 FPS Dmg2Former-AR 4×). At low resolutions (224 × 224), on the other hand, utilizing adaptive rescalers does not introduce efficiency improvements and may even slightly impact Dmg2Former speed, as seen in Table 8. Therefore, it is recommended to deploy adaptive rescaling when processing medium-to-higher resolution images.

## 5. Conclusions

As more progress is made in hardware and sensing technology, engineering inspectors gain access to an increasing amount of high-resolution data that would benefit from the advances in artificial intelligence. Fast and efficient visual inspection is preferred for real-time applications, but the resource demands of state-of-the-art vision transformer models are exponentially increasing. We proposed Dmg2Former-AR, a high-resolution visual inspection framework that encompasses a low-resolution transformer segmentation network, Dmg2Former, and two trainable resizers inspired by efficient subpixel convolution and Laplacian pyramid networks.

By testing our framework on the Material Segmentation dataset, we found an increase in IoU values and other evaluation metrics with progressive scaling using the proposed adaptive rescalers with factors of two, four, and eight. On the other hand, using interpolation-based resizers resulted in a deterioration of the model performance compared to the low-resolution segmentation model results. The differences in performance between the adaptive rescaling-based and interpolation-based models are more pronounced in the crack segmentation task. When inspecting the low-resolution output of the adaptive downsampler, uniform downsampler, and the segmentation model for both, we found that the use of adaptive rescalers in Dmg2Former-AR significantly helped in preserving the fine crack features, producing more accurate crack maps compared to the interpolation-based resizers. Furthermore, we inspected the computational demands for different configurations of Dmg2Former and found that upscaling Dmg2Former using adaptive rescalers offers a significantly efficient alternative to high-resolution segmentation.

Currently, this method only detects and maps cracks. However, there is a need to identify crack measurements, such as length and width, for some condition assessment tasks, such as alkali-silica reaction crack growth monitoring. Additionally, this is a supervised learning method which cannot generalize to other visual inspection tasks without being trained on such tasks. We anticipate that our approach can potentially predict masks for different damage types, such as spalling, rebar exposure, and steel rust, as long as they are trained on sufficient annotated data that include these classes. Finally, machine-learning models make mistakes, especially when attempting to extrapolate beyond their knowledge. Providing a measure of uncertainty as part of the given output can allow inspectors to take appropriate action when confidence drops. Uncertainty quantification is often implemented by formulating a posterior probability distribution (or an approximate) over the model parameters using Bayes’ theorem [11,29,79]. Implementing an uncertainty quantification strategy for damage segmentation in Dmg2Former-AR is recommended for future studies.

## Figures and Tables

**Figure 1 sensors-24-06007-f001:**
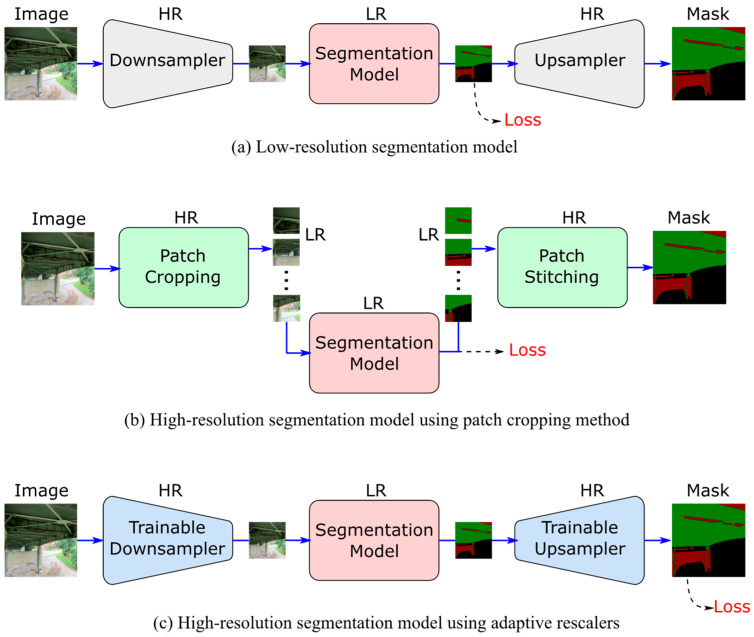
High-resolution segmentation approaches (HR: high resolution; LR: low resolution).

**Figure 2 sensors-24-06007-f002:**
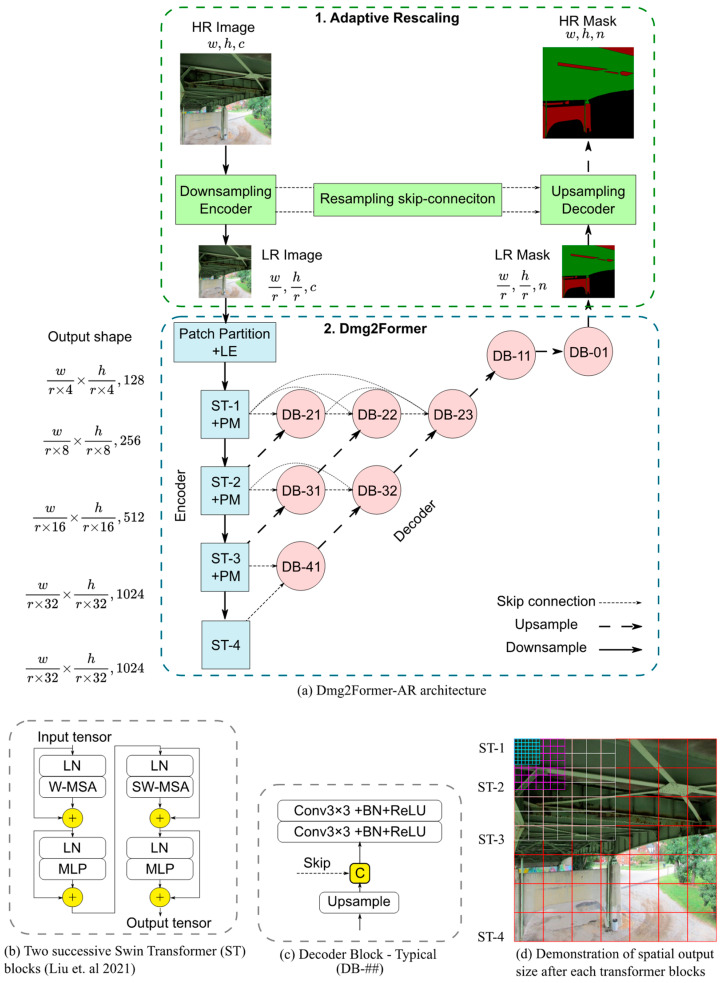
Dmg2Former-AR architecture. (*c*: channels, *n*: classes, ST-*i*: Swin Transformer block, D-*ij*: decoder block, Conv: 2D convolution, BN: batch normalization, ReLU: rectified linear unit, LN: layer normalization, MLP: multi-layer perceptron, SW/W-MSA: regular and shifted windowed multi-head self-attention, LE: linear embedding, PM: patch merging [52]).

**Figure 3 sensors-24-06007-f003:**
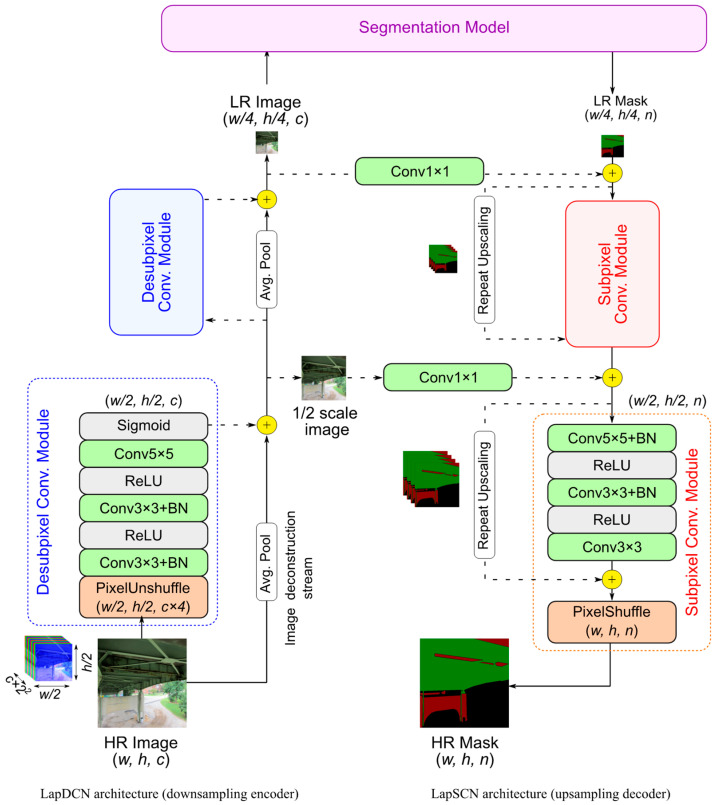
Downsampling and upsampling network architectures. (*c*: channels, *n*: classes, Conv: 2D convolution, BN: batch normalization, ReLU: rectified linear unit).

**Figure 4 sensors-24-06007-f004:**
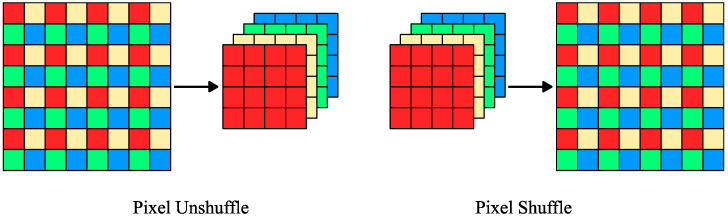
Pixel Shuffle and Pixel Unshuffle operations.

**Figure 5 sensors-24-06007-f005:**
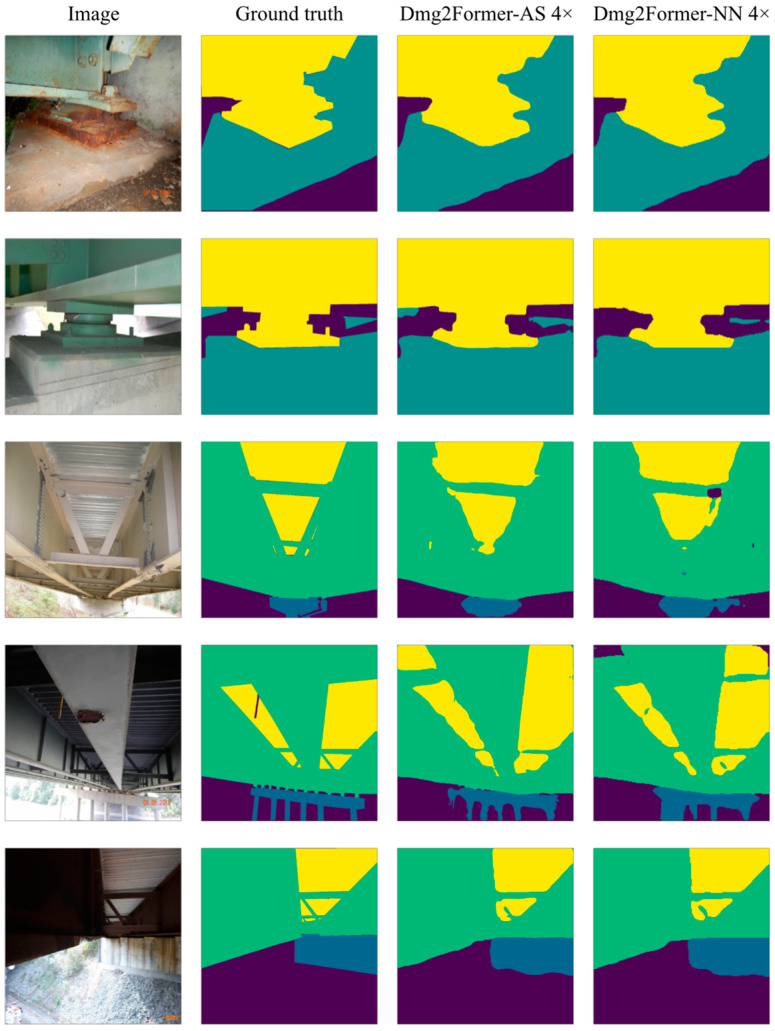
Material segmentation sample test images and their Dmg2Former-AR 4× and Dmg2Former-NN 4× predictions. Encoders were initialized with ImageNet-pre-trained weights.

**Figure 6 sensors-24-06007-f006:**
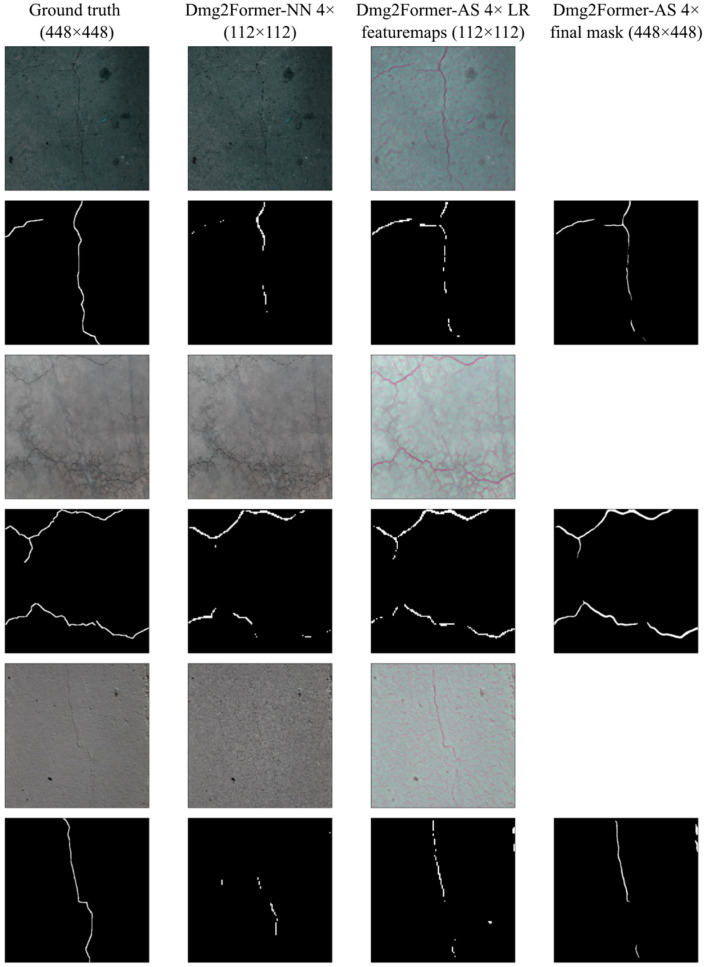
Insights into the Dmg2Former-AR inference process. From the left, the ground-truth images and masks containing cracks at 448 × 448, the output of nearest-neighbor and LapDCN downsampling images by a factor of four (top) and the corresponding masks of Dmg2Former at 112 × 112 (bottom), and the LapSCN-upscaled mask at 448 × 448.

**Figure 7 sensors-24-06007-f007:**
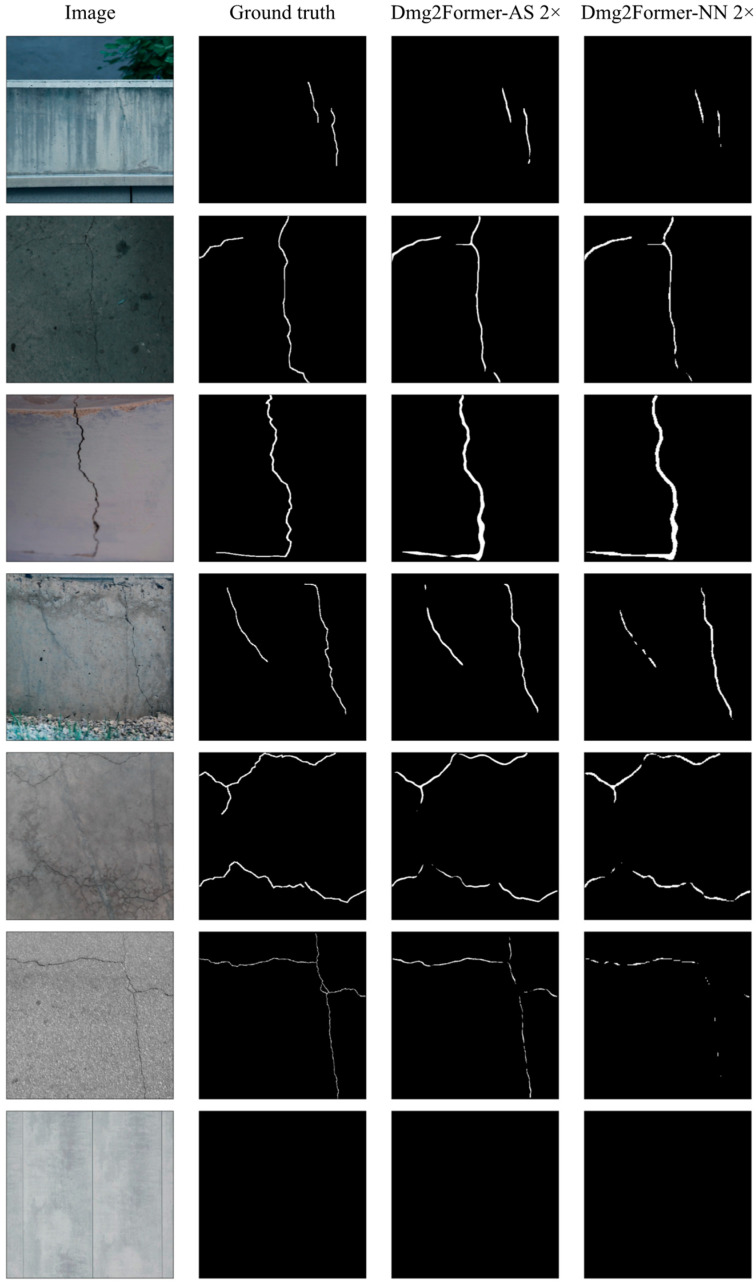
Crack segmentation sample test images and their Dmg2Former-AR 2× and Dmg2Former-NN 2× predictions. Encoders were initialized with ImageNet-pre-trained weights.

**Table 1 sensors-24-06007-t001:** DCN and UCN specification.

	Desubpixel Conv. Module (Downsampling)			Subpixel Conv. Module (Upsampling)		
Layer #	Operator	Kernel	# Channels	Operator	Kernel	# Channels
1	PixelUnshuffle	4 × 4	48	Conv.	5 × 5	64
2	Conv.	3 × 3	32	Conv.	3 × 3	32
3	Conv.	3 × 3	64	Conv.	3 × 3	*n* × 4^2^
4	Conv.	3 × 3	3	PixelShuffle	4 × 4	*n*

**Table 2 sensors-24-06007-t002:** Composition of the Concrete Crack Conglomerate dataset [68].

Dataset	Image Count
CrackForest Dataset [69]	118
Crack500 [70]	3363
Cracktree200 [71]	206
DeepCrack [72]	521
EugenMiller [70]	55
GAPs [70]	509
Rissbilder [70]	1411
Non-crack [70]	3822
Volker [70]	990
Total image count	10,995

**Table 3 sensors-24-06007-t003:** Materials segmentation testing performance metrics (average) grouped by output image size—randomly initialized parameters. The best-obtained metrics in a group are shown in bold font.

	Dmg2Former Size	Image Size	F1-Score (%)	IoU (%)	Recall (%)	Precision (%)
Dmg2Former	112	112	86.52	76.76	85.30	87.93
Dmg2Former-NN 2×	112	224	84.64	73.89	83.24	86.32
Dmg2Former-AR 2×	112	224	87.99	79.01	87.13	88.95
Dmg2Former	224	224	**88.21**	**79.36**	**87.16**	**89.41**
Dmg2Former-NN 4×	112	448	84.47	73.65	83.03	86.24
Dmg2Former-AR 4×	112	448	88.41	79.68	87.38	89.59
Dmg2Former-NN 2×	224	448	87.25	77.84	86.14	88.54
Dmg2Former-AR 2×	224	448	**89.29**	**81.09**	**88.47**	**90.26**
Dmg2Former-NN 8×	112	896	84.38	73.49	82.85	86.24
Dmg2Former-AR 8×	112	896	88.76	80.19	88.05	89.53
Dmg2Former-NN 4×	224	896	87.27	77.87	86.14	88.58
Dmg2Former-AR 4×	224	896	**89.15**	**80.85**	**88.36**	**90.04**

**Table 4 sensors-24-06007-t004:** Materials segmentation testing performance metrics (average). Encoders were initialized with ImageNet-pre-trained weights. The best-obtained metrics are shown in bold font.

	Dmg2Former Size	Image Size	F1-Score (%)	IoU (%)	Recall (%)	Precision (%)
Dmg2Former	224	224	92.18	85.80	92.15	92.26
Dmg2Former-NN 2×	224	448	91.32	84.35	90.77	91.96
Dmg2Former-NN 4×	224	896	91.25	84.23	90.65	91.94
Dmg2Former-AR 2×	224	448	**92.78**	**86.81**	**92.51**	93.11%
Dmg2Former-AR 4×	224	896	92.55	86.48	92.09	**93.19%**

**Table 5 sensors-24-06007-t005:** Materials segmentation class-wise IoU results. Encoders were initialized with ImageNet-pre-trained weights. The best-obtained metrics are shown in bold font.

	Image Size	Background	Concrete	Steel	Metal Decking
Dmg2Former	224	73.92	86.31	94.40	88.59
Dmg2Former-NN 2×	448	72.23	85.22	93.28	86.69
Dmg2Former-NN 4×	896	72.07	85.25	93.27	86.32
Dmg2Former-AR 2×	448	**75.18**	**87.06**	94.57	90.44
Dmg2Former-AR 4×	896	73.84	86.62	**94.78**	**90.69**

**Table 6 sensors-24-06007-t006:** Crack segmentation testing performance metrics (average) grouped by output image size—randomly initialized parameters. The best-obtained metrics in a group are shown in bold font.

	Dmg2Former Size	Image Size	F1-Score (%)	IoU (%)	Recall (%)	Precision (%)
Dmg2Former	112	112	70.49	54.43	68.86	72.19
Dmg2Former-NN 2×	112	224	69.29	53.01	68.41	70.18
Dmg2Former-AR 2×	112	224	**72.60**	**56.98**	**74.39**	70.89
Dmg2Former	224	224	72.09	56.35	70.95	**73.25**
Dmg2Former-NN 4×	112	448	68.20	51.75	67.18	69.26
Dmg2Former-AR 4×	112	448	73.62	58.26	74.08	73.17
Dmg2Former-NN 2×	224	448	71.45	55.58	69.66	73.34
Dmg2Former-AR 2×	224	**448**	**74.04**	**58.79**	**74.67**	**73.42**
Dmg2Former	448	448	73.32	57.88	74.04	72.61

**Table 7 sensors-24-06007-t007:** Crack segmentation testing performance metrics. Encoders were initiated with ImageNet-pre-trained weights. The best-obtained metrics are shown in bold font.

	Image Size	F1-Score (%)	IoU (%)	Recall (%)	Precision (%)
Dmg2Former	224	74.54	59.41	75.70	73.40
Dmg2Former-NN 2×	448	74.41	59.25	75.41	73.44
Dmg2Former-AR 2×	448	**76.07**	**61.39**	**78.29**	**73.98**

**Table 8 sensors-24-06007-t008:** Dmg2Former model statistics grouped by output image size—four classes output.

	Dmg2Former Size	Image Size	Trainable Params	MAC (G)	Forward Pass Size (MB)	Inference Speed (Frames-per-Second)
Dmg2Former	112	112	108, 987, 980	7.24	309.9	49.1
Dmg2Former-AR 2×	112	224	109, 044, 719	7.95	350.7	46.9
Dmg2Former	224	224	108, 992, 588	7.25	309.9	48.5
Dmg2Former-AR 4×	112	448	109, 101, 442	10.77	514.1	45.1
Dmg2Former-AR 2×	224	448	109, 049, 327	10.08	473.3	46.7
Dmg2Former	448	448	108, 992, 588	28.84	1239.6	33.6
Dmg2Former-AR 8×	112	896	109, 158, 165	22.08	1167.6	43.3
Dmg2Former-AR 4×	224	896	109, 106, 050	21.39	1126.8	45.5
Dmg2Former	896	896	108, 992, 588	115.17	4958.2	9.6

## Data Availability

The data are contained within the article.

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
