# Peer review of "Dmg2Former-AR: Vision Transformers with Adaptive Rescaling for High-Resolution Structural Visual Inspection"

_sensors, 2024, doi:10.3390/s24186007_

Round 1

Reviewer 1 Report

Comments and Suggestions for Authors

This study introduced a novel semantic segmentation architecture that combines vision transformers with Laplacian pyramid scaling networks to allow for rapid pixel-level damage identification in high-resolution visual inspection data. Resampling or cropping high-resolution images in visual damage detection models often causes the loss of critical fine details, such as thin cracks and edges, or essential global contextual features. The deep Laplacian pyramid networks are trained to rescale images non-uniformly, preserving vital inspection-related information within substantially smaller dimensions. This study also proposed an vision-transformer segmentation model that utilizes the saved computational overhead to output detailed visual inspection masks. Before the final publication of this study, the following issues should be addressed.

1.     The innovation of this study should be further emphasized in the abstract.

2.     The proposed method is validated through a series of experiments using publicly available visual inspection datasets for various tasks, including crack detection and material identification. Whether this method will cause anomaly misjudgment?

3.     To validate the proposed method, different kind of visual data were collected and used for different scenarios. For the monitoring data, whether the data preprocessing has been done?

4.     The research review can be further supplemented and improved, and it is suggested to supplement the literatures on damage or crack detection methods based other sensing technologies and machine learning method, e.g., RFID sensing techniques and Bayesian machine learning method, which can refer to: Towards long-transmission-distance and semi-active wireless strain sensing enabled by dual-interrogation-mode RFID technology, Review of wireless RFID strain sensing technology in structural health monitoring.

5.     The quality of some figures should be further improved.

6.     There exist some typos and syntax errors, and please double-checked the full manuscript.

7.     In conclusions, please provide some discussions on future research direction.

Comments on the Quality of English Language

Minor revision required.

Author Response

Please find attached response letter.

Reviewer 2 Report

Comments and Suggestions for Authors

In this paper, an adaptive multi-scale approach is used to solve the high-resolution image processing problem. The study is interesting. The suggested corrections are listed in the next paragraphs.

1. Based on the references in the manuscript, most of the cited research achievements come from more than two years ago. In fact, the machine vision technique to detect structural defects is developing rapidly. Suggest to investigate the latest research findings.

2. The contributions of the article should be further clarified.

3. Is the Dmg2Former model a new network architecture proposed in this study? How does it differ from the Swin Transformer? Suggest to provide more explanation in Section 2.2.

4. In practical applications, the real-time performance of structural surface damage detection is a key. How efficient is the detection of Dmg2Former-AR? In the case studies, whether it is material segmentation or crack segmentation, it seems that a detection efficiency analysis should be conducted.

5. In the case analysis, the two methods described in the second section, high-resolution segmentation model and high-resolution segmentation model using adaptive rescalers, should be used as comparative experiments. The corresponding calculation results should be provided. Additionally, a prediction efficiency analysis should be conducted.

Author Response

Please find attached response letter.

Reviewer 3 Report

Comments and Suggestions for Authors

This paper proposed an algorithm of vision transformers with adaptive rescaling for high-resolution structural visual inspection. The topic is meaningful and within the scope of Sensors. However, some technical details of the paper are not clear, and require further clarification. The main issues are listed below:

1. The Abstract is too tedious and needs to be refined and highlight innovations or contributions.

2. There are too many contents authors want to express in Figure 2, and the current version appears confusing and lacks readability. It is suggested dividing the relevant contents with wireframes, which may increase the readability of Figure 2.

3. In the reviewer's opinion, processes of pixel shuffle and pixel unstuffle are more suitable for scenarios from high pixels to low pixels. When the processes are applied to scenarios from low pixels to high pixels, it may introduce more errors. If the author also agrees, please add appropriate explanation for the discussion of Figure 4.

4. From Figure 5, it can be observed that the task of material segmentation may be influenced by structural surface defects. In the authors’ research, how to simultaneously complete material segmentation and defect feature (such as crack length and width) recognition without influencing each other.

5. This paper repeatedly mentions ‘adaptive’, but the reviewer does not find an effective way to achieve the so-called ‘adaptive’. The authors should add relevant discussions or change the wording in the text.

6. The Conclusions mention ‘high-dimensional data’, but in fact, the image is only two-dimensional data. It is recommended changing it to ‘high-resolution data’. Similar issues throughout the whole text should also be revised. Scientific papers should maintain objectivity and not overly exaggerate the functionality of the research content.

7. In the field inspection of bridges, the uncertainty of data and inspection process is the main factor limiting the expected effectiveness of artificial intelligence algorithms. The author should add some discussions on the robustness of uncertainty. Some relevant literature can be used as evidence, such as: https://doi.org/10.1109/TIM.2023.3343742

Comments on the Quality of English Language

The English of this paper needs a minor revision.

Author Response

Please find attached response letter.

Reviewer 4 Report

Comments and Suggestions for Authors

Summary:

This manuscript introduces a novel semantic segmentation architecture named Dmg2Former-AR, which integrates Vision Transformers and Laplacian pyramid scaling networks for high-resolution structural visual inspection. The study aims to address the time-consuming issue in traditional visual inspection when dealing with a large amount of image data, and to enhance the efficiency and reliability of the inspection by combining the latest computer vision and artificial intelligence technologies.

Major Issues:

1.        The paper primarily validates the model's performance on a specific dataset, but lacks an in-depth discussion on the model's generalization capabilities across different environments and types of structural damage. Further exploration into its robustness under various environments and conditions, as well as potential limitations, would be beneficial.

2.        The manuscript presents an innovative semantic segmentation architecture, Dmg2Former-AR, which combines Vision Transformers and Laplacian pyramid scaling networks for high-resolution visual inspection data. The authors claim that their approach can handle high-resolution images while maintaining computational efficiency. However, the authors do not provide a direct comparison with the state-of-the-art models in the field in terms of computational costs, such as training time or inference speed.

Minor Issues:

1.        The paper emphasizes the efficiency and accuracy of the proposed Dmg2Former-AR model in the conclusion section. However, the conclusion lacks a discussion on current limitations and future research directions.

Comments on the Quality of English Language

English language fine.

Author Response

Please find attached response letter.

Round 2

Reviewer 1 Report

Comments and Suggestions for Authors

No further comment.

Comments on the Quality of English Language

Minor editing of English language required.

Reviewer 2 Report

Comments and Suggestions for Authors

The authors take into account the comments of the review. The manuscript could be accepted in present form. 

Reviewer 3 Report

Comments and Suggestions for Authors

The authors have now made adequate changes and this paper can be accepted.

Comments on the Quality of English Language

The English of this paper needs a minor revision.